# Luteolin Induces Nrf2 Activity in C2C12 Cells: Implications for Muscle Health

**DOI:** 10.3390/ijms26094092

**Published:** 2025-04-25

**Authors:** Nicole Böttcher, Frank Suhr, Thomas Pufe, Christoph Jan Wruck, Athanassios Fragoulis

**Affiliations:** 1Department of Anatomy and Cell Biology, Uniklinik RWTH Aachen, 52074 Aachen, Germany; frank.suhr@uni-bayreuth.de (F.S.); tpufe@ukaachen.de (T.P.); cwruck@ukaachen.de (C.J.W.); afragoulis@ukaachen.de (A.F.); 2Division of Molecular Exercise Physiology, Faculty of Life Sciences: Food, Nutrition and Health, University of Bayreuth, 95326 Kulmbach, Germany

**Keywords:** Nrf2, Nrf2 activators, luteolin, C2C12 myoblasts, skeletal muscle, SkM

## Abstract

Chronic oxidative distress results in cellular damage, necessitating adaptive mechanisms for redox balance. The transcription factor nuclear factor erythroid 2-related factor 2 (Nrf2) is pivotal in the regulation of key antioxidant and cytoprotective genes. Under normal conditions, Nrf2 undergoes rapid degradation through polyubiquitination. However, it can be activated during oxidative eustress and distress via modifications of its inhibitor Kelch-like ECH-associated protein 1 (KEAP1). Activation of the Nrf2-Keap1 signaling pathway may decelerate aging-related muscle degeneration, such as sarcopenia and cachexia. In this study, we investigated the efficacy of two muscle-active endogenous factors, creatine and L-β-aminoisobutyric acid (L-BAIBA), as well as two natural phytochemicals, luteolin and silibinin, to induce Nrf2 in the murine myoblast cell line C2C12. Our results revealed that only luteolin significantly enhances Nrf2 activity in both proliferating and differentiated C2C12 cells, leading to increased expression of Nrf2 target genes in proliferating C2C12 cells. In contrast, the other three compounds had either no or only minor effects on Nrf2 activity or target gene expression. Our results underscore the distinct responses of C2C12 cells to different Nrf2 activators, emphasizing the significance of cellular context in their biological effects and highlight luteolin as a potential future treatment option to counteract muscle wasting associated with sarcopenia and cachexia.

## 1. Introduction

The chronic and excessive presence of oxidants can lead to cellular damage, a condition that is more accurately referred to as oxidative distress, rather than oxidative stress in general, as it is often referred to in the literature. Hence, mechanisms to maintain cellular redox homeostasis have evolved in mammalian cells. The transcription factor nuclear factor erythroid 2-related factor 2 (NF2L2 or Nrf2) plays a crucial role in this response by regulating the expression of genes such as glutathione peroxidase 2 (*GPX2*), heme oxygenase-1 (*HMOX1*) and NAD(P)H:quinone oxidoreductase 1 (*NQO1*). Under basal conditions, Nrf2 is rapidly degraded by the proteasome due to its polyubiquitination, mediated either by its canonical inhibitor Kelch-like ECH-associated protein 1 (KEAP1) [1] or alternatively by its non-canonical inhibitor beta-transducin repeats-containing proteins (β-TrCP) [2].

In the physiological context, a moderate amount of oxidants already leads to the activation of Nrf2 through oxidative modification of KEAP1. This cellular state, in which Nrf2 mainly performs its protective function and thus the amount of oxidants does not yet lead to cellular damage, is called oxidative eustress rather than oxidative distress. However, Nrf2 can be activated independently of oxidants by inhibiting the glycogen synthase kinase 3 beta (GSK-3β)/β-TrCP non-canonical pathway [2]. Once dissociated from KEAP1, Nrf2 is liberated to translocate into the nucleus, where it can heterodimerize with small musculoaponeurotic fibrosarcoma (sMAF) proteins to bind to antioxidant response elements (AREs) in the promoter and regulatory regions of Nrf2 target genes [1].

Emerging evidence suggests that Nrf2 also influences skeletal muscle (SkM) metabolism and health [3]. Under physiological conditions, the expression and activation of Nrf2 in SkM are strictly regulated [4,5]. However, disturbances in the Nrf2-Keap1 signaling pathway may contribute to age-related SkM damage via oxidative distress [6], including conditions such as sarcopenia and cachexia [7,8,9]. Sarcopenia is defined as the age-related loss of SkM mass and function, commonly observed in elderly patients. It leads to decreased mobility, increased risk of falls and an overall decline in quality of life [10]. However, sarcopenia can also occur in patients with other conditions, such as cancer, obesity or rheumatic diseases [11]. It results from a complex interplay of pathophysiological factors, including enhanced inflammation, dysregulated signaling pathways and increased oxidative distress. For instance, a study using a cancer cachexia model indicates that Nrf2 functions as a key player in protecting against SkM atrophy [12].

Numerous pharmacological Nrf2 activators have been identified, primarily electrophilic compounds that modify cysteine residues on KEAP1. One notable example is the isothiocyanate sulforaphane (SFN), which targets cysteine 151 of KEAP1 to release Nrf2 through electrophilic modification [13]. Other activators include protein–protein interaction inhibitors, like the small molecule Cpd15 [14], or natural compounds such as silibinin, the biologically active component of silymarin (*Silybum marianum*) [15,16]. Fundamental in silico studies of binding poses and affinity for several compounds of silymarin revealed that they are very likely able to perform a disruption of the Keap1-Nrf2 complex [17]. Modulation of the AKT/GSK3β/Nrf2/GPX4 pathway is another potential mechanistic effect of silymarin as an Nrf2 activator [18]. Additionally, the naturally occurring flavonoid luteolin has been demonstrated to activate the Nrf2 pathway [19,20,21]. This occurs through the increased phosphorylation of Nrf2 induced by MEK1/2-ERK1/2, which in turn increases Nrf2′s stability and favors a conformational change of Keap1 so that translocation into the nucleus is favored [19,22]. Furthermore, muscle-active endogenous factors such as L-β-aminoisobutyric acid (L-BAIBA), a metabolite of the branched-chain amino acid (BCAA) metabolism, promote the translocation of Nrf2 into the nucleus by activating the adenosine monophosphate-activated protein kinase (AMPK) signaling pathway through increased phosphorylation of AMPK [23]. Joo and colleagues reported that the Ser550 residue of Nrf2 is phosphorylated by AMPK and, in this way, promotes the dissociation of Nrf2 from Keap1 [24]. Interestingly, creatine monohydrate—one of the most popular supplements for muscle gain—seems to possess antioxidant properties due to an activation of the Nrf2 pathway [25]. In a similar way to L-BAIBA, creatine has also been shown to increase the levels of p-AMPK [26]. However, some proposed Nrf2 activators exhibit ambivalent effects, acting as potential inhibitors under certain conditions, such as luteolin [27]. These observations highlight that Nrf2 activators demonstrate high variability regarding their potential to activate Nrf2, depending on the tested cell system and the respective cell states. Given the significance of Nrf2 in SkM, the characterization/validation of the described Nrf2 activators in proliferating and differentiating myogenic precursors is of high relevance to further advance pharmacological strategies of Nrf2 regulation.

In this study, we applied four distinct Nrf2 activators—luteolin, silibinin, creatine and L-BAIBA—in the C2C12 myoblast in vitro model [28] to identify candidates that effectively activate Nrf2 in both proliferating myoblasts as well as differentiated myocytes. As there is still limited data on their efficacy in muscle tissue, these four candidates were selected to represent the two different groups of natural and endogenous compounds to be tested. Nrf2 activation has the potential to alleviate SkM loss by inhibiting the inflammatory pathway and promoting cellular health. Our research aims to identify suitable treatment approaches to activate Nrf2 and thereby promote muscle health in future in vivo studies.

## 2. Results

### 2.1. Luteolin Is the Most Potent of the Tested Nrf2 Activators in Proliferating C2C12 Cells

In order to identify the ideal concentrations of the chosen Nrf2 activators (luteolin, creatine, L-BAIBA and silibinin) with minimal impact on cell viability in proliferating cells, we conducted CellTiter-Blue^®^ Cell Viability Assays (CTB). There was no detectable decrease in cell viability of proliferating C2C12 cells after treatment with luteolin in concentrations up to 12.5 µM. However, the use of 25 µM luteolin decreased relative cell viability compared to control, but without reaching statistical significance. A significant reduction in cell viability was evident at the highest concentration tested, 50 µM [57.09 ± 2.84% vs. control: 100 ± 8.50%] (Figure 1A). Similarly, a comparable influence on cell viability was observed for the two highest concentrations of creatine, 25 mM and 50 mM [25 mM: 80.08 ± 5.28%, 50 mM: 56.31 ± 5.50% vs. control: 100 ± 11.83%] (Figure 1B). The effects of silibinin were less gradual; both the vehicle control using DMSO at a dilution level of 1% and the group treated with 200 µM silibinin [DMSO: 66.0 ± 7.78%, 200 µM: 34.29 ± 3.25% vs. control: 100 ± 8.69%] exhibited a significant reduction in viability after 24 h incubation (Figure 1C). Consequently, the application of 200 µM silibinin on cells is not feasible due to the requirement for a high DMSO concentration of 1%, which may adversely affect cell viability.

Lastly, L-BAIBA showed statistically significant effects on cell viability after 24 h, starting at a concentration of 25 µM, with subsequent concentrations of 50, 100 and 200 µM also resulting in significant reductions in cell viability [25 µM: 92.83 ± 6.0%; 50 µM: 91.55 ± 5.65%; 100 µM: 92.41 ± 6.40%; 200 µM: 92.19 ± 3.15% vs. control: 100 ± 4.69%] (Figure 1D). Although these effects are likely biologically less relevant.

We identified the highest concentrations of the substances that did not adversely affect cell viability or significantly deviate from control values as the most suitable for subsequent experiments (indicated by the red dotted frames in Figure 1). Their potency to activate Nrf2 activity was analyzed with an antioxidant response element (ARE)-driven luciferase reporter gene assay. The stimulation periods of 8 and 24 h were chosen to analyze short-term and long-term effects. The 8 h stimulation allows us to observe acute responses, while the 24 h stimulation enables us to assess sustained changes, providing a comprehensive understanding of the effects of the activators.

Following stimulation with luteolin, an increase in ARE activity was observed after 24 h of treatment with the two highest concentrations, 6.25 µM and 12.5 µM [6.25 µM: 2.83 ± 0.85 x-fold; 12.5 µM: 5.79 ± 0.56 x-fold vs. control: 1.0 ± 0.37 x-fold]. However, no effects were detectable after 8 h of luteolin treatment (Figure 2A). Creatine exhibited no impact on ARE activity at both time points across all three concentrations evaluated (Figure 2B). The treatment with 10 µM silibinin for 8 h resulted in a significant elevation of ARE activity [1.20 ± 0.11 x-fold vs. control: 1.0 ± 0.1 x-fold]. A comparable effect was observed after 24 h of treatment with both 10 µM and 50 µM silibinin [10 µM: 1.45 ± 0.23 x-fold; 50 µM: 1.44 ± 0.12 x-fold vs. control: 1.0 ± 0.19 x-fold] (Figure 2C). Notably, a minor but significant increase in ARE activity was induced by the stimulation with 200 µM L-BAIBA for only 8 h [1.20 ± 0.10 x-fold vs. control: 1.0 ± 0.05 x-fold]. However, this effect was no longer detectable after a stimulation period of 24 h (Figure 2D).

To further verify these results, we investigated the gene expression of known Nrf2 target genes, in particular *Nqo1* and *Hmox1*, in proliferating C2C12 cells using RT-qPCR (Figure 3). These two genes were selected as representative examples of genes that have an antioxidant or anti-inflammatory impact [29,30].

Our results indicate that *Nqo1* expression significantly increased after 24 h of stimulation with luteolin at a concentration of 3.125 µM [1.74 ± 0.23 x-fold vs. control: 1.0 ± 0.27 x-fold] compared to the untreated control group (Figure 3A). Aside from the treatment with 50 µM silibinin, which also resulted in a significant enhancement of *Nqo1* expression [1.38 ± 0.20 x-fold vs. control: 1.0 ± 0.27 x-fold] (Figure 3C), none of the other treatments affected *Nqo1* gene expression (Figure 3B,D).

Regarding the Nrf2 target gene *Hmox1*, its expression was exclusively elevated after 24 h of stimulation with luteolin (Figure 3A) at concentrations of either 6.25 or 12.5 µM [6.25 µM: 1.43 ± 0.16 x-fold; 12.5 µM: 2.98 ± 0.65 x-fold vs. control: 1.0 ± 0.28 x-fold]. None of the other tested Nrf2 activators had any effect on *Hmox1* expression levels (Figure 3B–D).

### 2.2. Luteolin Is the Only Substance Tested That Activates Nrf2 in Differentiated C2C12 Cells

The proliferation of C2C12 cells induces notable changes in both morphology and protein expression, which consequently influence their sensitivity to various Nrf2 activators. To investigate this hypothesis, we aimed to determine whether differentiated C2C12 cells exhibit a different sensitivity to these Nrf2 activators compared to proliferating C2C12 cells. Cell viability was assessed using a CTB assay.

Our results indicate that differentiated C2C12 cells were sensitive to luteolin concentrations ranging from 3.125 µM to 50 µM. There was a significant reduction in the viability of cells [3.125 µM: 87.08 ± 9.01%; 6.25 µM: 78.28 ± 6.38%; 12.5 µM: 87.66 ± 7.38%; 25 µM: 81.59 ± 7.77%; 50 µM: 81.30 ± 8.69% vs. control: 100 ± 10.07%] (Figure 4A).

Additionally, differentiated C2C12 cells displayed increased sensitivity to creatine. With the exception of the concentration of 3.125 mM, significant decreases in cell viability were observed across the other concentrations tested [1.5 mM: 90.60 ± 6.24%; 6.25 mM: 75.24 ± 7.44%; 12.5 mM: 77.63 ± 5.19%; 25 mM: 66.95 ± 12.22%; 50 mM: 54.40 ± 17.38% vs. control: 100 ± 7.93%]. However, it is important to note that the vehicle group with H_2_O also resulted in decreased viability, indicating that the observed effects cannot be solely attributed to creatine (Figure 4B). Treatment with silibinin for 24 h revealed no impact on cell viability even at the highest tested concentration of 200 µM (Figure 4C). Interestingly, treatment with L-BAIBA resulted in a significant increase in viable cells at both concentrations of 1 µM and 100 µM [1 µM: 112.6 ± 8.90%; 100 µM: 116.6 ± 5.83% vs. control: 100 ± 7.43%] (Figure 4D). These findings highlight the differential responses of differentiated versus proliferating C2C12 cells to various Nrf2 activators and underscore the importance of cellular context in evaluating their biological effects.

Following the determination of appropriate concentrations for differentiated C2C12 cells, as indicated by the red dotted frames in Figure 4, Nrf2 activation was analyzed via ARE-driven luciferase reporter gene assay as before (Figure 5).

None of the tested concentrations of luteolin resulted in increased Nrf2 activity after stimulating C2C12 cells for 8 h. However, after 24 h of stimulation with both 6.25 µM and 12.5 µM luteolin, a significant increase in ARE activity was observed [6.25 µM: 1.98 ± 0.23 x-fold, 12.5 µM: 3.5 ± 0.94 x-fold vs. control: 1.0 ± 0.22 x-fold] (Figure 5A).

In contrast, the other three Nrf2 activators tested exhibited no substantial difference in ARE activity across the evaluated concentrations and time points (Figure 5B–D). These findings emphasize the varying effects of different Nrf2 activators and highlight luteolin’s potential to effectively modulate this pathway in the applied in vitro model for SkM research. The potential impact of Nrf2 activity on differentiated cells was further investigated through gene expression analyses of *Nqo1* and *Hmox1* using RT-qPCR (Figure 6).

No significant differences were observed for *Nqo1* expression after 24 h of stimulation across the investigated luteolin, creatine or L-BAIBA when compared to the control group (Figure 6A,B,D). Similarly, *Hmox1* expression also showed no statistically significant variations among the different treatments with luteolin, creatine or L-BAIBA compared to the control group. Interestingly, after 24-h stimulation with 50 µM silibinin, a significantly increased gene expression of both *Nqo1* and *Hmox1* was observed [*Nqo1*: 1.62 ± 0.25 x-fold vs. control 1.0 ± 0.26 x-fold; *Hmox1*: 1.59 ± 0.53 x-fold vs. control 1.0 ± 0.19 x-fold] (Figure 6C). These findings suggest that luteolin does not modulate the expression of either *Nqo1* or *Hmox1* under the conditions tested. Instead, silibinin significantly modulates gene expression.

## 3. Discussion

Oxidative distress, especially if sustained, has a significant molecular and functional impact on SkM, particularly in pathological conditions such as sarcopenia [31]. Activating the Nrf2 pathway represents a promising strategy for treating various pathologies, particularly as a potential adjuvant therapy option for patients with muscle-wasting disorders, such as sarcopenia. Sarcopenia can be addressed through interventions such as regular resistance training and nutritional strategies [32], including the use of omega-3 fatty acids combined with resistance training to enhance muscle isometric strength [33]. However, these interventions encounter considerable challenges, including issues of accessibility, individual variability in responses and physical factors [34]. Beyond that, there are currently no specific medications approved for the treatment of sarcopenia [10].

To improve the functional outcome for these patients, an adjunctive pharmacological approach targeting the Nrf2 pathway may be beneficial. Activation of Nrf2 enhances antioxidant defenses by upregulating genes responsible for producing antioxidant enzymes, thereby maintaining redox homeostasis and preventing oxidative distress known to be associated with muscle aging [35]. One of these genes is *Nqo1*, which is crucial for the detoxification of quinones [36]. It is also recognized for its ability to bind, stabilize and protect its target proteins from proteasomal degradation [37,38]. Moreover, since chronic inflammation exacerbates sarcopenia, activation of the Nrf2 pathway could contribute to a significant reduction in inflammatory markers. Notably, Nrf2 plays a pivotal role in regulating the expression of *Hmox1*, a potent anti-inflammatory enzyme [39,40]. Additionally, Nrf2 is known to inhibit NF-ΚB-mediated expression of pro-inflammatory genes [41]. Its role further extends to cellular repair processes, potentially aiding recovery and preserving SkM tissue during periods of disuse or injury-induced immobility. In particular, Wang et al. demonstrated that sulforaphane-induced Nrf2 activation improved grip strength and restored muscle fiber organization in a mouse model of diabetes [42].

The investigation of the optimal Nrf2 activators in both proliferating and differentiated C2C12 cells provided valuable insights into their efficacy and potential applications. Our findings demonstrated that luteolin is a potent Nrf2 activator in both cell states, supporting previous studies that emphasize its role as an Nrf2 inducer [19,43,44,45]. Notably, Zhang et al. demonstrated that luteolin treatment can mitigate SkM mass loss in a diet-induced obesity model [46]. However, there is also contradictory evidence that luteolin may inhibit Nrf2 signaling under certain circumstances, particularly in cancer models [27,47]. This contradiction highlights the complexity of luteolin’s effects, which appear to be context-dependent and can vary significantly between normal physiological conditions and pathological states such as cancer. A potential application strategy could be to induce the hormetic effect that Nrf2 activation can provide [48] through treatment with luteolin. Elmazoglu and colleagues have already described such an effect of luteolin in this manner in a study involving rotenone-induced toxicity in microglial cells [49]. With regard to prospective applications of luteolin in in vivo studies or clinical trials, it is important to consider administration in terms of dietary sources, doses and bioavailability. As a flavonoid, luteolin sources are abundant in various dietary supplements, such as broccoli (*Brassica oleracea*), sage (*Salvia officinalis*) or fenugreek seed (*Trigonella foenum-graecum*) [50,51]. In an in vivo model of periodontal disease in rats, high doses of luteolin (100 mg/kg) have been reported to be beneficial [52]. However, as these doses of luteolin are difficult to achieve from plant sources, additional supplementation would be advisable. When choosing the right supplement, it is important to keep in mind that flavonoids in powder form have a low absorption rate by the small intestine [53]. The solution for luteolin is a liposomal formulation that is available in a wide range of products [54]. Further investigation is required to elucidate these mechanisms and to clarify the conditions under which luteolin exerts either activating or inhibiting effects on the Nrf2 pathway.

Evidence from a study by Tie and colleagues suggests that, for silibinin to exert its full potential, a pathological trigger must first be applied, and silibinin acts as a rescue mechanism [55]. Our findings align with this, showing that, while silibinin modestly activates Nrf2 in proliferating C2C12 myoblasts, it does not induce Nrf2 in differentiated cells. Interestingly, a significant increase in the gene expression of the two Nrf2 target genes, *Hmox1* and *Nqo1*, was detected in differentiated C2C12 cells. This increase is approximately 1.5-fold, and it remains uncertain whether this would be sufficient for biological relevance in an in vivo model. This shows that there may be a possible underlying potential for silibinin to induce Nrf2 activation in both proliferating and differentiated cells under different circumstances. Overall, it appears to be less potent than luteolin. The lack of prior oxidative challenge before treatment with silibinin may account for its relatively subdued effects [55]. Similar results were observed in a cisplatin-induced oxidative stress mouse model of muscle atrophy [56]. This underscores the importance of cellular context and preconditioning in maximizing the therapeutic potential of Nrf2 activators.

Previous studies on L-BAIBA have shown that it can induce Nrf2 signaling and thereby significantly reduce ferroptosis in a mouse model of ischemia–reperfusion-induced lung injury [23]. However, our data demonstrated that L-BAIBA had a very limited effect on Nrf2 activity in proliferating C2C12 myoblasts, and only at relatively high concentrations. This discrepancy implies that L-BAIBA may exert a more pronounced influence on Nrf2 signaling under pathological conditions compared to the in vitro environment we studied. Another possible reason might be a tissue-specific effect of L-BAIBA.

We made a similar observation with respect to creatine. While the existing literature suggests that creatine combined with physical exercise leads to elevated levels of Nrf2 in a Parkinson’s disease mouse model [26], our research demonstrated that creatine supplementation alone did not lead to significant induction of the Nrf2 signaling pathway in C2C12 myoblasts. This raises an intriguing question about the role of physical activity as a potential enhancer of creatine’s effectiveness as an Nrf2 modulator. Further exploration of this relationship may provide valuable insights into how lifestyle factors influence the efficacy of dietary supplements on cellular signaling pathways.

While our study provides valuable insights into the effects of various Nrf2 activators, it is not without limitations. First, it is important to note that our research relies on an in vitro cell culture model. Future studies incorporating in vivo data will be crucial for validating and extending our findings to more complex biological contexts. Additionally, the fixed time points selected for our experiments may not capture the full range of effects. Certain responses could potentially manifest at different time intervals. Moreover, in differentiated cells cultured at a high confluence, the conditions may inadvertently induce oxidative stress and elevate Nrf2 levels independently of treatment effects. This could complicate our ability to discern the specific contributions of each Nrf2 activator under these circumstances. Therefore, these factors must be considered when interpreting our results and highlight the necessity for further investigation to fully understand the dynamics of Nrf2 modulation.

In conclusion, our study identified distinct profiles for different Nrf2 activators regarding their modulation of antioxidant response element (ARE) activity and Nrf2 target gene expression in C2C12 cells. The inability to validate several proposed Nrf2 activators from the literature highlights the necessity for context-specific assessments of their functionality. Luteolin emerges as a particularly promising candidate due to its ability to activate protective genes. Future research should focus on elucidating the molecular mechanisms underlying these responses and exploring their therapeutic implications for oxidative stress-related muscle diseases. One potential in vivo approach may involve combining Nrf2 activators, such as luteolin, with existing therapeutic options to achieve synergistic benefits that target multiple pathways involved in muscle-wasting diseases such as sarcopenia and cachexia. While current strategies are essential for treating sarcopenia, incorporating Nrf2 activation could enhance outcomes by reducing oxidative distress and damage, as well as inflammation, ultimately promoting better muscle health in older adults.

## 4. Materials and Methods

### 4.1. Cell Culture

The C2C12 myoblast cell line was utilized for this study [28]. For the ARE-driven luciferase reporter gene assays, cells were transduced with a SIN-lentiARE construct, as described previously [57]. Successfully transduced cells were selected using 2 µg/mL puromycin (Carl Roth GmbH + Co. KG, Karlsruhe, Germany, #0240.4). Proliferating C2C12 cells were cultivated with Dulbecco’s Modified Eagle’s Medium (DMEM) high glucose (Sigma-Aldrich, St. Louis, MO, USA, #D5796-500ML) containing 10% fetal calf serum (FCS) (Life Technologies Limited, Paisley, UK, #10500-064) and 1% penicillin/streptomycin (Sigma-Aldrich, St. Louis, MO, USA, #P0781-100ML), ensuring a confluence of no more than 75% to prevent spontaneous differentiation. To induce differentiation, the medium was exchanged to DMEM containing 2% horse serum (PAA Laboratories GmbH, Pasching, Austria, #B15-122) and 1% penicillin/streptomycin at 80% confluence. Cells were monitored microscopically on a regular basis for myotube development. After an average of 4 days, the cells were differentiated (see Appendix A).

### 4.2. Stimulation

Nrf2 activators used included creatine monohydrate (Merck, Darmstadt, Germany, #C3630), L-BAIBA (Merck, Darmstadt, Germany, #72713-10MG), luteolin (Merck, Darmstadt, Germany, #440025) and silibinin (LKT Laboratories, St. Paul, MN, USA, #S3343). Each compound was dissolved in either dimethylsulfoxid (DMSO) (Carl Roth GmbH + Co. KG, Karlsruhe, Germany, #A994.2) or MilliQ H_2_O as appropriate. Vehicle controls matched the solvent used for each stimulant at the highest concentration.

### 4.3. Cell Viability Assay

To investigate the effects of luteolin, creatine, silibinin and L-BAIBA on cell viability six different concentrations per stimulant were tested: luteolin (1.5, 3.125, 6.25, 12.5 and 25 µM), creatine (1.5, 3.125, 6.25, 12.5 and 25 mM), and both silibinin and L-BAIBA at concentrations of 1, 10, 25, 50, 100 and 200 µM. Cells were stimulated for a duration of 24 h before viability assessment using the CellTiter-Blue^®^ (CTB) Viability Assay (Promega, Madison, WI, USA, #G8080), in accordance with the manufacturer’s instructions. Assessments were conducted with an Infinite^®^ M200 Plate Reader (TECAN, Männedorf, Switzerland).

### 4.4. ARE Activity

ARE activity was quantified using a luciferase reporter gene assay. At the end of the stimulation (8 or 24 h), the cells were washed with PBS to remove residual culture medium. Afterwards, the cells were resuspended in 1X passive lysis buffer (part of the applied luciferase assay system; Promega, Madison, WI, USA, #E1501). Lysis was performed by initial shaking for 20 min at 500 rpm on a plate agitator at room temperature, followed by three freeze–thaw cycles (−80 °C to 4 °C on ice) for optimal efficiency. The lysates obtained were used both for the measurement of ARE-dependent luciferase activity, according to the manufacturer’s recommendations (Promega, Madison, WI, USA, #E1501), and for the measurement of total DNA using the CyQuant™ Cell Proliferation Assay (Invitrogen, Waltham, MA, USA, #C7026), for the purpose of data normalization.

### 4.5. RNA Isolation, cDNA Synthesis and qPCR

The RT-qPCR study was conducted in accordance with the MIQE guidelines [58]. RNA was isolated from 300,000 cell/6-well using RNA-Solv^®^ Reagent (Omega Bio-tek, Norcross, GA, USA, #R6830-02), following the manufacturer’s protocol. Nucleic acid quantity was examined spectrophotometrically using a NanoDrop^®^ND-1000 (Thermo Scientific, Waltham, MA, USA). RNA purity was determined using A_260_/A_280_ and A_260_/A_230_ ratios of the samples. RNA integrity was verified by MOPS-buffered denaturizing agarose gel electrophoresis (see Appendix A). For reverse transcription of 2 µg total RNA, Maxima™ Reverse Transcriptase was utilized (Thermo Fisher Scientific, Waltham, MA, USA, #EP0743), in combination with mixed priming (3:1, *v*/*v*) of oligo-(dT)18:random hexamer (Thermo Fisher Scientific, Waltham, MA, USA, #SO132 and #SO142). Real-time PCR was performed on a QuantStudio™ 3 system (Thermo Fisher Scientific, Waltham, MA, USA) using PowerSYBR^®^Green PCR Master Mix (Thermo Fisher Scientific, Waltham, MA, USA, #4367660). The design of primers, featuring binding sites for the included transcripts, is illustrated in Appendix A. Primer-specific, pre-evaluated annealing temperatures and a specific cycle protocol were applied (see Appendix A). All qPCR primer and their specific annealing temperatures are listed in Appendix A. Primer specificity was confirmed by melt-curve analysis and TAE-buffered DNA agarose gel electrophoresis of the PCR product (see Appendix A). Potential variability between PCR runs was corrected using inter-run calibrators.

Amplification efficiency was evaluated with LinRegPCR 2017.0 software (Heart Failure Research Center, Amsterdam, The Netherlands) based on Ramakers et al.’s methodology [59]. A reference gene index for normalization was established using the geNorm algorithm from a variety of potential reference genes, identifying the combination of succinate dehydrogenase complex subunit A (*Sdha*) and cullin 4A (*Cul4a*) for proliferating and differentiated C2C12 cells as optimal. All tested reference gene primer sequences, annealing temperatures and additional relevant information can be found in our previous study on reference gene selection [60]. Relative fold-change in gene expression was calculated using the efficiency-corrected ΔΔCq method via qbase+ v3.4 (CellCarta, Montreal, QU, Canada) as described by Pfaffl [61].

### 4.6. Statistics

Data were analyzed by JMP 10.0.0 (SAS Institute Inc., Cary, NC, USA) and GraphPad Prism 10.4.1 (Graphpad Software Inc., La Jolla, CA, USA). Normal distribution of data was tested with the Shapiro–Wilk test and variance homogeneity through Bartlett’s test. BoxCox-Y or Log10 transformation was conducted to achieve homoscedasticity if necessary (indicated for respective data sets). One-way ANOVA and Dunnett’s multiple comparisons test were applied for parametric data, while non-parametric data were analyzed using the Kruskal–Wallis-test followed by Dunn’s multiple comparisons test. Statistical significance was defined as * *p* < 0.05, ** *p* < 0.01 and *** *p* < 0.005. Data are represented as arithmetic mean + standard deviation (SD).

### 4.7. Graphic Illustration

Graphical abstract: Created in BioRender. Böttcher, N. (2025) https://BioRender.com/dx5l7nb.

## Figures and Tables

**Figure 1 ijms-26-04092-f001:**
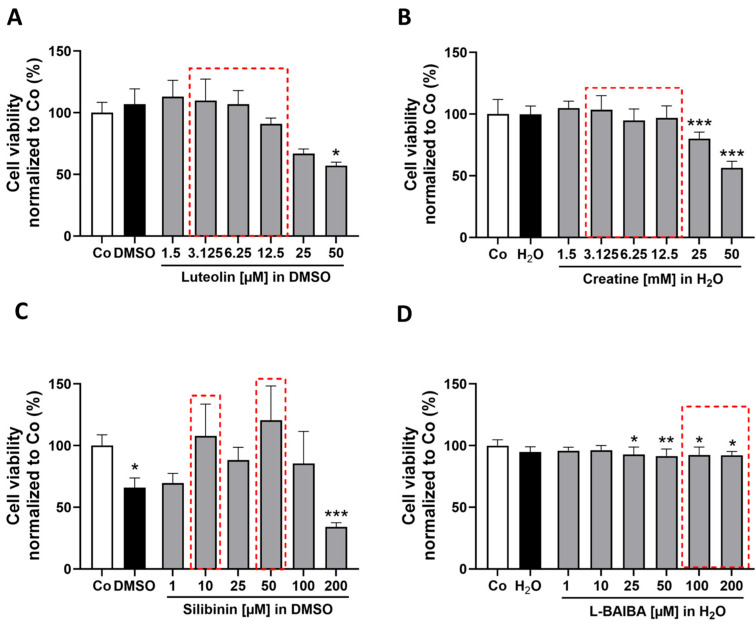
Analysis of cell viability in proliferating cells following 24 h exposure to Nrf2 activators. Cell viability was evaluated for four previously characterized Nrf2 activators at varying concentrations using the CTB assay. (**A**) Luteolin [1.5, 3.125, 6.25, 12.5, 25 and 50 µM], (**B**) creatine [1.5, 3.125, 6.25, 12.5, 25 and 50 mM], (**C**) silibinin [1, 10, 25, 50, 100 and 200 µM] and (**D**) L-BAIBA [1, 10, 25, 50, 100 and 200 µM]. Control groups (Co) consisted of cell culture medium along with the appropriate vehicle controls (DMSO or MilliQ H_2_O). Data are represented as mean + SD with n = 6. Statistical significance is indicated as * *p* < 0.05, ** *p* < 0.01 and *** *p* < 0.005 compared to the control group. Analyses were performed using either a one-way ANOVA and subsequent Dunnett’s multiple comparisons post hoc test (**B**,**D**) or a Kruskal–Wallis test and subsequent Dunn’s multiple comparisons post hoc test (**A**,**C**). The red dotted outlines highlight the selected concentrations of the stimulants for the subsequent experiments.

**Figure 2 ijms-26-04092-f002:**
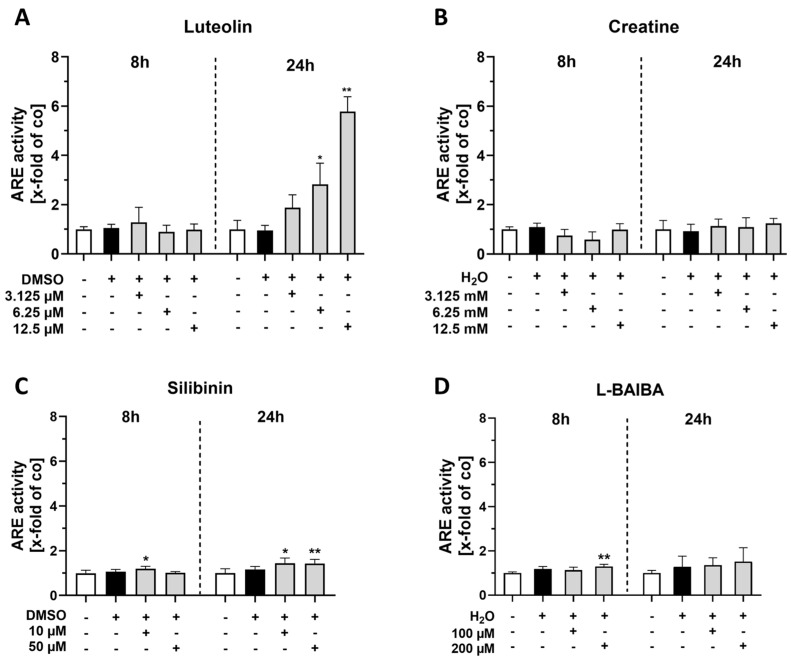
ARE luciferase reporter gene assays of proliferating C2C12 cells after 8 or 24 h of Nrf2 activator stimulation. The activity of ARE was evaluated for the previously determined concentrations of the four Nrf2 activators: (**A**) Luteolin [3.125, 6.25 and 12.5 µM], (**B**) creatine [3.125, 6.25 and 12.5 mM], (**C**) silibinin [10 µM and 50 µM] and (**D**) L-BAIBA [100 and 200 µM]. Control groups (Co) included cell culture medium and corresponding vehicle controls (DMSO or MilliQ H_2_O). Data were normalized to DNA amount in the samples using a CyQuant™ Cell Proliferation assay. Data are represented as mean + SD with n = 6 for all groups with the exception of Control and DMSO groups in (**C**) with n = 12. Statistical significance was determined using a Kruskal–Wallis test and subsequent Dunn’s multiple comparisons post hoc test, with significance levels indicated as * *p* < 0.05 and ** *p* < 0.01 compared to the untreated control group.

**Figure 3 ijms-26-04092-f003:**
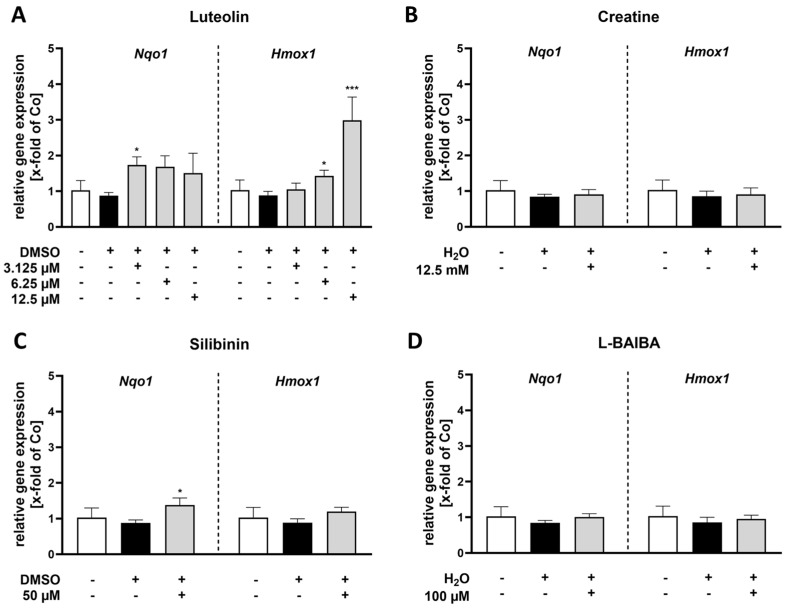
RT-qPCR analysis of Nrf2 activity markers in proliferating C2C12 cells following 24 h stimulation with Nrf2 activators. Relative gene expression as x-fold of the control group (Co) of either *Nqo1* or *Hmox1* is represented for (**A**) luteolin, (**B**) creatine, (**C**) silibinin and (**D**) L-BAIBA. Data were normalized to the untreated control and represented as mean + SD with n = 6. Statistical significance is indicated as * *p* < 0.05 and *** *p* < 0.005 compared to the control group. A one-way ANOVA followed by a Dunnett’s multiple comparisons post hoc test was applied for (**A**–**D**), while a Kruskal–Wallis test with a subsequent Dunn’s multiple comparisons post hoc test was utilized in (**A**) for *Nqo1* gene expression. For the *Nqo1* data sets in (**B**,**D**), a Box-Cox-Y transformation was applied, while a log10 transformation was used for the *Hmox1* data set in (**A**) prior to statistical analysis.

**Figure 4 ijms-26-04092-f004:**
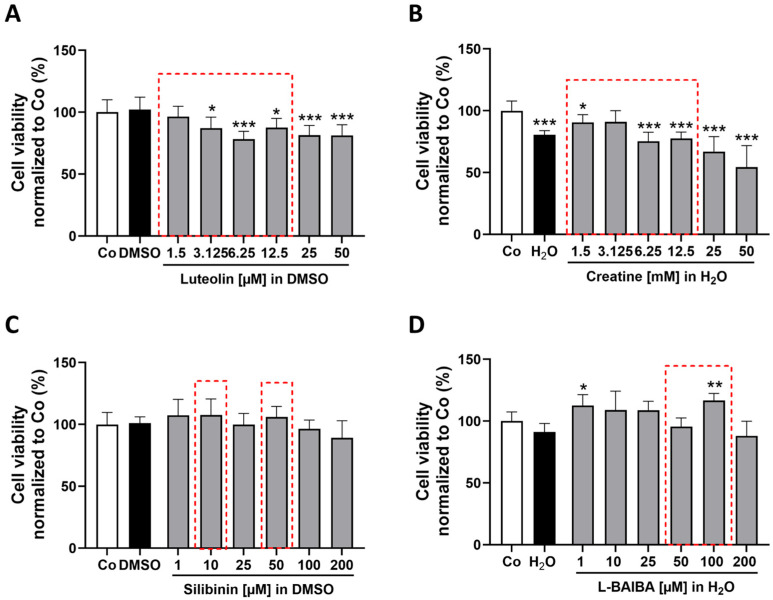
Assessment of cell viability in differentiated C2C12 cells following 24 h stimulation with Nrf2 activators. Cell viability was investigated for the four different previously described Nrf2 activators at varying concentrations using the CTB assay: (**A**) Luteolin [1.5, 3.125, 6.25, 12.5, 25 and 50 µM], (**B**) creatine [1.5, 3.125, 6.25, 12.5, 25 and 50 mM], (**C**) silibinin [1, 10, 25, 50, 100 and 200 µM] and (**D**) L-BAIBA [1, 10, 25, 50, 100 and 200 µM]. Control groups (Co) consisted of cell culture medium along with the corresponding vehicle controls (DMSO or MilliQ H_2_O). Data are represented as mean + SD with n = 6. Statistical significance is indicated as * *p* < 0.05, ** *p* < 0.01 and *** *p* < 0.005 compared to the control group. Analyses were performed using a one-way ANOVA followed by a Dunnett’s multiple comparisons post hoc test. For the data sets (**B**) and (**D**), a Box-Cox-Y transformation was performed prior to statistical analysis. The selected concentrations of the stimulants used are highlighted by red dashed boxes.

**Figure 5 ijms-26-04092-f005:**
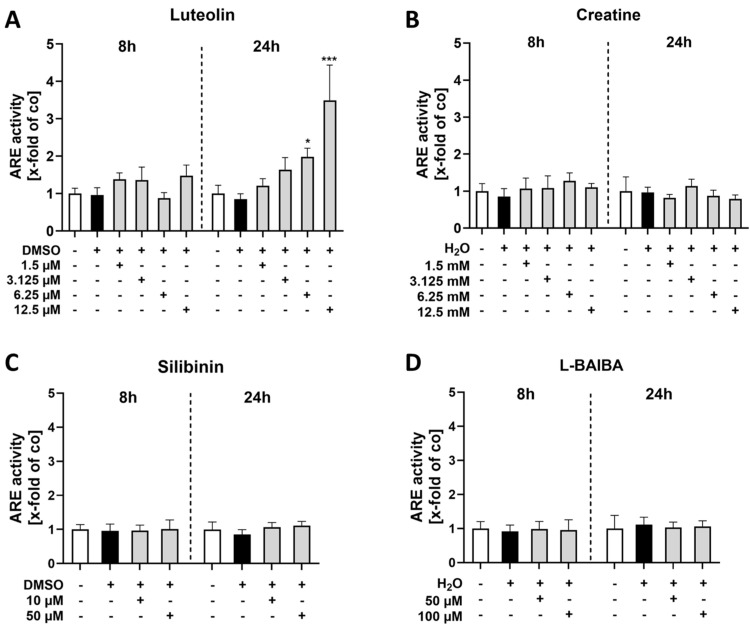
ARE luciferase reporter gene assays of differentiated C2C12 cells after 8 or 24 h of Nrf2 activator stimulation. ARE activity in C2C12 cells was evaluated for the four Nrf2 activators at varying concentrations: (**A**) luteolin [1.5, 3.125, 6.25 and 12.5 µM], (**B**) creatine [1.5, 3.125, 6.25 and 12.5 mM], (**C**) silibinin [10 µM and 50 µM] and (**D**) L-BAIBA [50 and 100 µM]. Each assay included an untreated control group (Co) consisting of cell culture medium and the corresponding vehicle controls (DMSO or MilliQ H_2_O). The data was normalized to cell number using a CyQuant assay and is represented as mean + SD with n = 6. Statistical significance was determined using a Kruskal–Wallis test with a subsequent Dunn’s multiple comparisons post hoc test, with significance levels indicated as * *p* < 0.05 and *** *p* < 0.005 compared to the control group.

**Figure 6 ijms-26-04092-f006:**
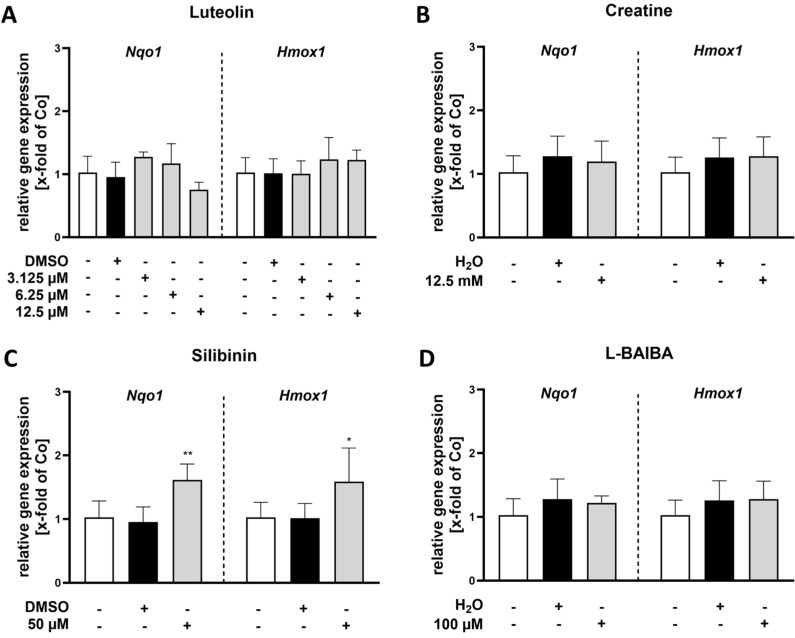
RT-qPCR analysis of Nrf2 activity markers in differentiated C2C12 cells following 24 h of stimulation with Nrf2 activators. Relative gene expression as x-fold of the control group (Co) of either *Nqo1* or *Hmox1* is represented for (**A**) luteolin, (**B**) creatine, (**C**) silibinin and (**D**) L-BAIBA. Data were normalized to the untreated control and represented as mean + SD with n = 6, with the exception for silibinin (*Nqo1*) n = 5. Statistical significance is indicated as * *p* < 0.05 and ** *p* < 0.01 compared to the control group. A one-way ANOVA followed by a Dunnett’s multiple comparisons post hoc test was applied for (**A**–**D**). For the *Nqo1* data set in (**A**) a log10 transformation and in (**B**) a Box-Cox-Y transformation was performed prior to statistical analysis.

## Data Availability

The original contributions presented in this study are included in the article/Appendix A. Further inquiries can be directed to the corresponding author.

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
