# Peer review of "Luteolin Induces Nrf2 Activity in C2C12 Cells: Implications for Muscle Health"

_ijms, 2025, doi:10.3390/ijms26094092_

Round 1
Reviewer 1 Report
Comments and Suggestions for Authors
Major Comments
1. In introduction section, although the rationale for selecting Luteolin, Silibinin, Creatine, and L-BAIBA is briefly addressed, the scientific basis for comparing these four compounds within a single experimental framework remains unclear particularly given the fundamental differences between endogenous metabolites (Creatine, L-BAIBA) and plant-derived phytochemicals (Luteolin, Silibinin). Moreover, the introduction lacks mechanistic insight into how each compound is expected to activate Nrf2. The claim that Nrf2 activation alone may counteract complex conditions such as cachexia appears to be an oversimplification, as cachexia involves multifactorial pathophysiology including cancer and chronic inflammation. Additionally, the Nrf2-inducing effects of luteolin have already been documented in previous studies, yet the novelty and added value of the current study are not clearly articulated.
5. The overall width of the error bars throughout the experiments appears to be quite large. The relatively large error bars observed across the experiments may be a cause for concern in terms of data consistency and variability.
6. In the discussion section, a more comprehensive examination of the underlying mechanisms, such as KEAP1 modification and Nrf2 nuclear translocation, is warranted, along with a clearer acknowledgment of the study’s limitations and potential directions for future research. Although the manuscript briefly notes that luteolin may inhibit Nrf2 under certain conditions, it lacks an in-depth analysis of the factors that might account for these contradictory effects. Additionally, the interpretation of results for silibinin, creatine, and L-BAIBA is limited, focusing primarily on the lack of observed effects. Comparing the present experimental conditions with previous studies where these compounds activated Nrf2 would help provide possible explanations for the discrepancies observed.
7. The current discussion seems to generalize in vitro findings to in vivo or even clinical applications. In particular, the conclusion that luteolin may serve as a promising therapeutic agent for sarcopenia or cachexia appears to be an overinterpretation based solely on cell-based experiments.
Minor Comments
1. When silibinin was treated to C2C12 cells, the cell viability was higher at 50 μM than at 25 μM. What could be the reason for this observation? Was this trend consistently observed across all three independent experiments? Additionally, further explanation is needed regarding the rationale for selecting 8 h and 24 h as the time points.
2. Why were only Nqo1 and Hmox1 chosen as Nrf2 target genes? Given that the Nrf2 pathway regulates a broad range of target genes, the rationale for selecting only these two genes is not clearly presented. Moreover, including descriptions of the physiological roles and functions of these genes would help improve the reader’s understanding.
3. When luteolin was applied to C2C12 cells, the viability of proliferating cells was lower than that of differentiated cells. This suggests that proliferating cells may be more sensitive to luteolin. Nevertheless, the manuscript states that “Differentiated C2C12 cells displayed a greater sensitivity to luteolin,” which appears contradictory. A more detailed explanation is needed to justify this interpretation.
4. In the RT-qPCR analysis, all four Nrf2 activators were tested in proliferating cells, but only luteolin was examined in differentiated cells. The reason for this discrepancy is not clearly explained. This experimental design makes direct comparisons difficult, and further clarification is required.
5. Overall, the number of experiments conducted appears to be limited. In particular, the claim that "luteolin activates Nrf2" seems to rely primarily on the single result showing that among the four tested activators, only luteolin significantly increased ARE activity. Therefore, the novelty of this study compared to previous research is unclear. Furthermore, the lack of explanation regarding why the same experimental conditions were not applied to both proliferating and differentiated cells weakens the overall completeness and persuasiveness of the manuscript.
6. Although the expression of Nrf2 target genes Nqo1 and Hmox1 significantly increased following luteolin treatment, the study lacks a detailed explanation or analysis of the reasons behind the differential expression observed between proliferating and differentiated cells.
7. In Figure 1C, the relatively large error bars may raise concerns about the precision and reproducibility of the experimental results.
8. In Figure 1D, significance markers (*, **) are shown for 25 µM, 50 µM, 100 µM, and 200 µM L-BAIBA treatments, yet the cell viability remains above 90% in all cases. The actual decrease in viability appears minimal, and the criteria used to determine statistical significance and its biological relevance may appear questionable.
9. The ARE luciferase reporter gene assays were limited to 8 and 24 hours, and clarification of the rationale behind the selection of these specific time points would be beneficial.
The English could be improved to more clearly express the research
Reviewer 2 Report
Comments and Suggestions for Authors
The manuscript addresses the important topic of Nrf2 activation in the context of muscle protection against oxidative stress and age-related degeneration. The study clearly defines the research problem, hypothesis and objectives, and the discussion provides an in-depth analysis of previous studies.
To enrich the manuscript, it is recommended that the molecular mechanisms of action of luteolin be investigated in more detail. In addition, a visual representation of luteolin's effects with graphic illustrations could improve the clarity and impact of the article.
The conclusion regarding the synergistic benefits of combining luteolin with resistance training and dietary interventions is not directly supported by the research presented. To substantiate this claim, mathematical modeling software should be used to determine the potential synergy based on computational models. Moreover, in terms of the practical application of luteolin's effects, a brief paragraph on dietary sources, intake doses and bioavailability of the molecule should be provided.
Round 2
Reviewer 1 Report
Comments and Suggestions for Authors
There are no comments other than comments on question 3.
3. When luteolin was applied to C2C12 cells, the viability of proliferating cells was lower than that of differentiated cells. This suggests that proliferating cells may be more sensitive to luteolin. Nevertheless, the manuscript states that “Differentiated C2C12 cells displayed a greater sensitivity to luteolin,” which appears contradictory. A more detailed explanation is needed to justify this interpretation.
A: It is true that the proliferating cells at higher concentrations of 25 and 50 µM have an average that is slightly lower than that of the differentiated cells. However, the average at the values of 1.5 to 12.5 µM is even above or at the same level as the control group. This is not the case with the differentiated C2C12 cells, which is why an increased sensitivity to luteolin has been mentioned.
R: The average at the values of 1.5 to 12.5 µM is even above or at the same level as the control group. -> However, the cell viability of proliferating cells treated with luteolin did not show a statistically significant difference compared to the control group within the tested concentration range. Furthermore, at higher concentrations, proliferating cells appeared to be more sensitive to luteolin, and this difference was statistically significant. Therefore, the sentence "Our results indicate that differentiated C2C12 cells displayed a greater sensitivity to luteolin in contrast to their proliferating counterparts" seems unnecessary and could be omitted for clarity.
Author Response
We would like to thank reviewer 1 for taking the time to review this manuscript. Please find below the detailed response to your comment.
C:3.When luteolin was applied to C2C12 cells, the viability of proliferating cells was lower than that of differentiated cells. This suggests that proliferating cells may be more sensitive to luteolin. Nevertheless, the manuscript states that “Differentiated C2C12 cells displayed a greater sensitivity to luteolin,” which appears contradictory. A more detailed explanation is needed to justify this interpretation.
A: It is true that the proliferating cells at higher concentrations of 25 and 50 µM have an average that is slightly lower than that of the differentiated cells. However, the average at the values of 1.5 to 12.5 µM is even above or at the same level as the control group. This is not the case with the differentiated C2C12 cells, which is why an increased sensitivity to luteolin has been mentioned.
R: The average at the values of 1.5 to 12.5 µM is even above or at the same level as the control group. -> However, the cell viability of proliferating cells treated with luteolin did not show a statistically significant difference compared to the control group within the tested concentration range. Furthermore, at higher concentrations, proliferating cells appeared to be more sensitive to luteolin, and this difference was statistically significant. Therefore, the sentence "Our results indicate that differentiated C2C12 cells displayed a greater sensitivity to luteolin in contrast to their proliferating counterparts" seems unnecessary and could be omitted for clarity.
A:We do agree that this sentence is possibly too imprecise. For this reason, it has been removed. The passage has been reworded accordingly.
“Our results indicate that differentiated C2C12 cells were sensitive to luteolin concentrations ranging from 3.125 µM to 50 µM. There was a significant reduction in the viability of cells [3.125 µM: 87.08 ± 9.01%; 6.25 µM: 78.28 ± 6.38%; 12.5 µM 87.66 ± 7.38%; 25 µM 81.59 ± 7.77%; 50 µM: 81.30 ± 8.69 % vs. control: 100 ± 10.07 %] (Figure 4 A).“